# Seismic response of a mid-story isolated stilted structure in mountainous areas

Jianhua Li[1], Dewen Liu [2]*

1 School of Civil Engineering & Architecture, Wenzhou Polytechnic, Wenzhou, China, 2 Krirk University, Bangkok, Thailand

* civil_liudewen@sina.com

**Data Availability Statement:** All the raw data needed for the research results (minimal data set) can be accessed via this data link. Datasets used in the research—Compressed files archive. https://doi.org/10.6084/m9.figshare.26877514.v1.

## Abstract

Research on the SSI effect on flat sites has yielded many valuable conclusions. However, current research on the impacts of various special local terrains on structural dynamics remains limited. For mountainous areas, it is common to construct houses in a multi-step, climbing, and laterally staggered architectural form that follows the mountain terrain. Only through the analysis of the combined action of the upper and lower parts can the seismic performance of this type of structural form be better revealed; considering the influence of SSI effects will be closer to the actual seismic effects. Therefore, to identify the damage factors of the mid-story isolated stilted structures under earthquakes and provide optimized design plans for the structures, six models are established considering three slopes and two types of foundations based on the engineering case in Chongqing, China. Through the elastic-plastic time-history analysis under earthquakes in the down and transverse-slope directions, concludes, compared with not considering SSI, the seismic response of the mid-story isolated stilted structures considering SSI in mountainous areas is amplified. With the increase of the mountain slope, the seismic response of the structures considering SSI increases, and the amplification coefficients are between 1–1.8. The amplification coefficients of the structures without SSI are concentrated around 1, which is less influenced by the slope. The damage to the stilted isolated layer is mainly concentrated in the column and the beam end, and the maximum seismic response appears in the short columns. The foundation soil stress increases with the increase of the mountain slope.

## 1 Introduction

There are many mountainous terrains in the world, research on the SSI effect on flat sites has yielded many valuable conclusions. However, current research on the impacts of various special local terrains on structural dynamics remains limited. For mountainous areas, it is common to construct houses in a multi-step, climbing, and laterally staggered architectural form that follows the mountain terrain. Only through the analysis of the combined action of the upper and lower parts can the seismic performance of this type of structural form be better revealed. For example, there are many mountainous terrains in southwest China. Additionally, it is an earthquake-prone area. Therefore, research on seismic mitigation and isolation for

**Funding:** The author(s) received no specific funding for this work.

**Competing interests:** The author(s) declared no potential conflicts of interest with respect to the research, authorship, and/or publication of this article.

mountainous buildings is particularly crucial. The montane stilted structure in HongYaDong, Chongqing, China, as shown in Fig 1, which is a type of structure adapted to the mountainous terrain and widely used worldwide. So the mid-story isolated stilted structures, which are affected by mountainous terrain, become research hotspot. The mid-story isolated stilted structures are stilted structures that adopt mid-story isolation. Mid-story isolated structure is a new type of isolated structure developed from base isolated structure, in which the isolation device is installed between the upper and lower stories of a building or set up between the sub-structure and the main structure, to control the earthquake response of the structure.

For the study of stilted structures, Yang Shijun [1] makes a push-over analysis on the stilted frame, and the result shows that the seismic response of the stilted structure in down-slope direction is more serious. Gong Guoqin and Xu Ge [2] elaborated on the design details and precautions of reinforced concrete stilted structures using practical engineering examples and analyzed their weak positions. Cheng et al. [3] investigated the failure probability of stilted structure by damaging the columns and how to reduce this failure probability under earthquakes by improving the flexural stiffness of columns. Welsh-huggins et al. [4], Liu et al. [5] and Liu, L. et al. [6] analyzed the seismic response of a hilly stilted-frame structure, and the result shows that the damage of a hilly stilted-frame structure is greater than that of a normal flat-ground frame structure. Li Yingmin et al. [7] found that the short columns connected to the ground significantly affected the stilted structure by analyzing its seismic response. For the study of the mid-story isolated structures, Li Aiqun et al. [8], Li Dali et al. [9], Song Xiao et al. [10] and Zhang et al. [11] conducted a large number of the elastic-plastic time-history analysis on the mid-story isolated structures, and the superior performance of mid-story isolated structures was confirmed in the analysis; and a semi-analytical method (SAM) for the mid-story isolated structures was proposed and its accuracy was verified by finite element simulation (FESE). Bolvardi et al. [12] developed a simple and easy-to-understand design method for displacement that determines the isolated layer parameters based on the desired performance. When seismic isolators are installed, the acceleration, displacement and inter-layer shear force of the structure can be significantly reduced, and the seismic performance gradually decreases as the isolated layer is positioned higher in the structure.

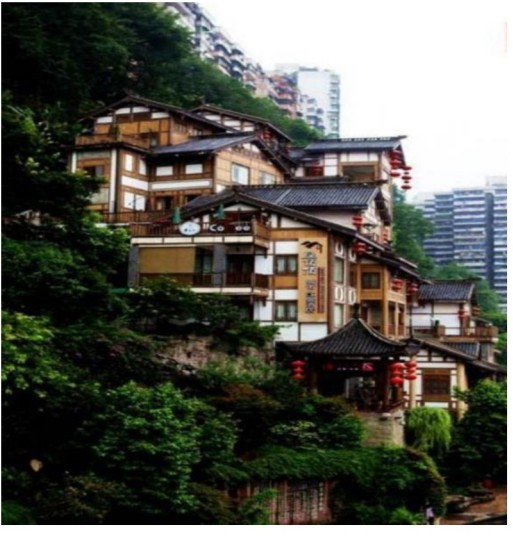 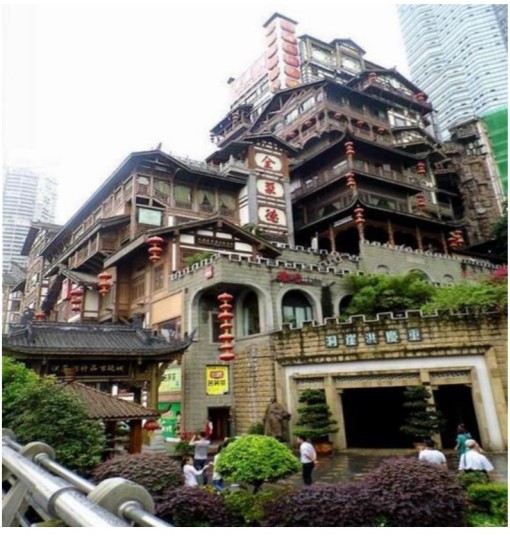

(a)                                                                                 (b)

**Fig 1. The mountanic stilted structure in HongYaDong, Chongqing.**

Soil-structure interaction (SSI) is a hot research topic both domestically and internationally at present, and the traditional assumption of rigid foundations has certain limitations [13–15]. Many scholars have mentioned in their studies that the soil-structure interaction has a strong influence on the superstructure [16–18]. Wu Yingxiong et al. [19] conducted a numerical simulation and shaking table test study of the mid-story isolated structure with large chassis and a single tower on rigid and soft interlayer grounds. The SSI effect can amplify the seismic response of the mid-story isolated structure and the failure probability of the structure. Mehdi et al. [20] used FLAC-2D finite element analysis software to model buildings of different heights under the assumption of infinite soil conditions to consider the soil-structure interaction. Yang et al. [21] analyzed the seismic response of energy dissipation devices under soil-structure interaction through studying the ground motion characteristics of near faults. Amicia L et al. [22] developed a modeling method to evaluate the comprehensive impact of various physical properties of the foundation on seismic velocity and anisotropy as the physical properties of the underlying rock have a great influence on the seismic response of the structure. Han Liutao et al. [23] considered the SSI effect of shaft towers in Class II and III sites and the results show that the acceleration of shaft towers was amplified approximately twofold.

Previous studies of mid-story isolated stilted structures often assumed rigid foundations, thereby ignoring the influence of the foundation on the structures. This is inconsistent with the actual engineering situation. SSI has an important influence on the dynamic characteristics of buildings, leading to an extension of the natural vibration period of the structure, making the natural vibration frequency of the superstructure close to the predominant period of the site, and then resonance may occur, resulting in more serious seismic damage phenomena in the superstructure. In mountainous areas, accounting for SSI effects more accurately reflects actual seismic impacts. Therefore, this paper established the mid-story isolated stilted structure models considering SSI. Three types of mountain slopes were considered. The paper conducts elastic-plastic time-history analyses of the structures under earthquakes in both down-slope and transverse-slope directions. Additionally, it establishes mid-story isolated stilted structure models without considering SSI for comparative analysis.

## 2 Structural theory analysis

In conventional seismic response analysis, the soil is usually considered to be a rigid body and the influence of soil on the dynamic response of the structure is not considered. In fact, because the foundation is not absolutely rigid body, there is not only the forces interaction, but also the mutual limitation of deformation between the structure and the foundation, which leads to the mutual propagation and exchange of vibration energy, so the dynamic response of the actual structure is very different from the dynamic response under the assumption of rigid foundation. In mountainous areas, the SSI effect has its special characteristics, and the soil's height and properties on either side of the structure may vary, so the seismic response analysis of mountainous buildings considering SSI has practical significance. The schematic diagrams of the mid-story isolated stilted structures in mountainous areas without SSI and with SSI considered are shown in Fig 2.

## 3 The establishment of finite element model

### 3.1 Project overview

A 10-story frame structure with a rectangular plan of 24 m ×24 m, the height of the ground floor is 4.5 m and the height of the other floors is 3.6 m. According to the Code for Seismic Design of Buildings (GB 50011–2010) [24]. The seismic precautionary intensity is VIII, the design basic seismic acceleration value is 0.20 g, the construction site classification is II, the

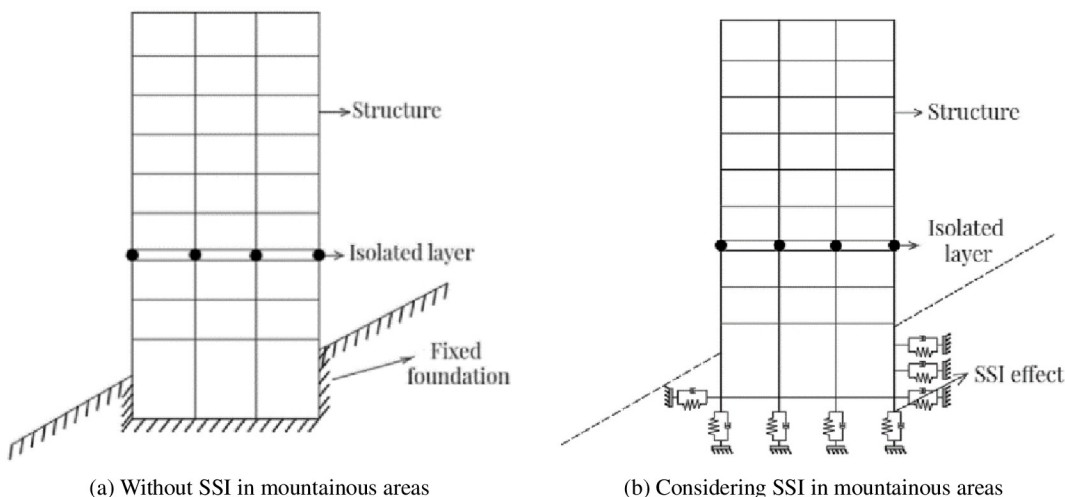

(a) Without SSI in mountainous areas          (b) Considering SSI in mountainous areas

**Fig 2. The schematic diagram of two different foundations.**

design seismic category is second group, and the isolated layer is set at the bottom of the column on the third floor. The concrete strength grade of the beam, column and slab is C40, and the concrete strength grade of the raft foundation is C30, the type of reinforced bar is HRB400, and the type of stirrup is HPB300, the thickness of the protection layer is 40 mm for the beam and slab and 60 mm for the column. Among them, the specified yield strength of HRB400 steel was $f_y = 400MP_a$; the specified yield strength of HPB300 steel was $f_y = 300MP_a$. Detailed parameters of the structure are shown in Table 1. The planar calculation sketch is shown in Fig 3.

## 3.2 The mid-story isolated stilted structure models considering SSI in mountainous areas

The mid-story isolated stilted structure models considering SSI in mountainous areas and without SSI were established, as shown in Fig 4. The structural frame was simulated by beam units and the floor slab was simulated by membrane units. Concrete and solid units adopt Takeda hysteretic model, while steel bars adopt Kinematic hysteretic model. The soil was simulated by solid elements and taking 10 times the size of the structure as the size of the solid unit, with the size of 240 m ×240 m. The soil was embedded in rigid bedrock and simulated using fixed hinge supports. The bottom of the frame adopts raft foundation and was embedded in the entity. Damper unit was used to simulate viscoelastic artificial boundary to absorb ground motions at the soil edge [25]. In the structure, the isolation bearing adopts isolation lead core rubber bearing (LRB). The models are LRB800 and LRB900. The detailed parameters

**Table 1. Parameters of the structure.**

| Component type | Floor | Cross-sectional dimensions (depth×width) | The thickness of protection layer for steel bar (mm) | Stirrup diameter (mm) | Reinforcement |
|---|---|---|---|---|---|
| Frame column | 1~10 | 600×600 | 60 | 10 | 16 C 25 |
| Main beam | 1~10 | 700×350 | 40 | 10 | 5 C 20 |
| Secondary beam | 1~10 | 600×300 | 40 | 10 | 5 C 20 |

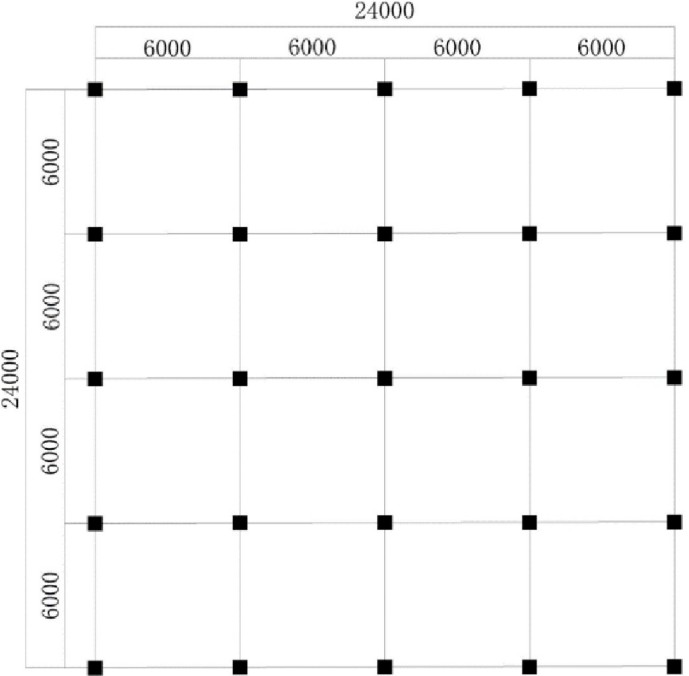

**Fig 3. Planar calculation sketch.**

of isolated bearings are shown in Table 2. The parameters of foundation soil are shown in Table 3.

### 3.3 Selection of seismic motions

In this paper, seven ground motions including EL Centro ground motion, Hollister ground motion, Kobe ground motion, San Fernando ground motion, Tangshan ground motion and two artificial ground motions numbered Ren-1 and Ren-2 are selected. The ground motions are input separately in down-slope direction and transverse-slope directions. According to the Code for Seismic Design of Buildings (GB 50011–2010) [24], and the research of Onur Araz and Chengqing Liu et al. [26–29], when using the time-history analysis method, actual strong earthquake records and artificially simulated acceleration time-history curves should be selected according to the building site category and design earthquake grouping. Specifically, the number of actual strong earthquake records should not be less than two-thirds of the total. The average seismic influence coefficient curve of multiple sets of time-history curves should be statistically consistent with the seismic influence coefficient curve used in the modal decomposition response spectrum method. The spectral amplitude under rare earthquakes is adjusted to 400 cm/s$^2$. The acceleration response spectrums of ground motions are shown in Fig 5, and the detailed parameters of seven ground motions are shown in Table 4.

## 4 Seismic response of structure

### 4.1 Comparative analysis of structural periods

Modal analysis was carried out for the mid-story isolated stilted structure considering SSI and without SSI under different mountain slopes. The first three structural periods are shown in Table 5, the period of the mid-story isolated stilted structure considering SSI in mountainous

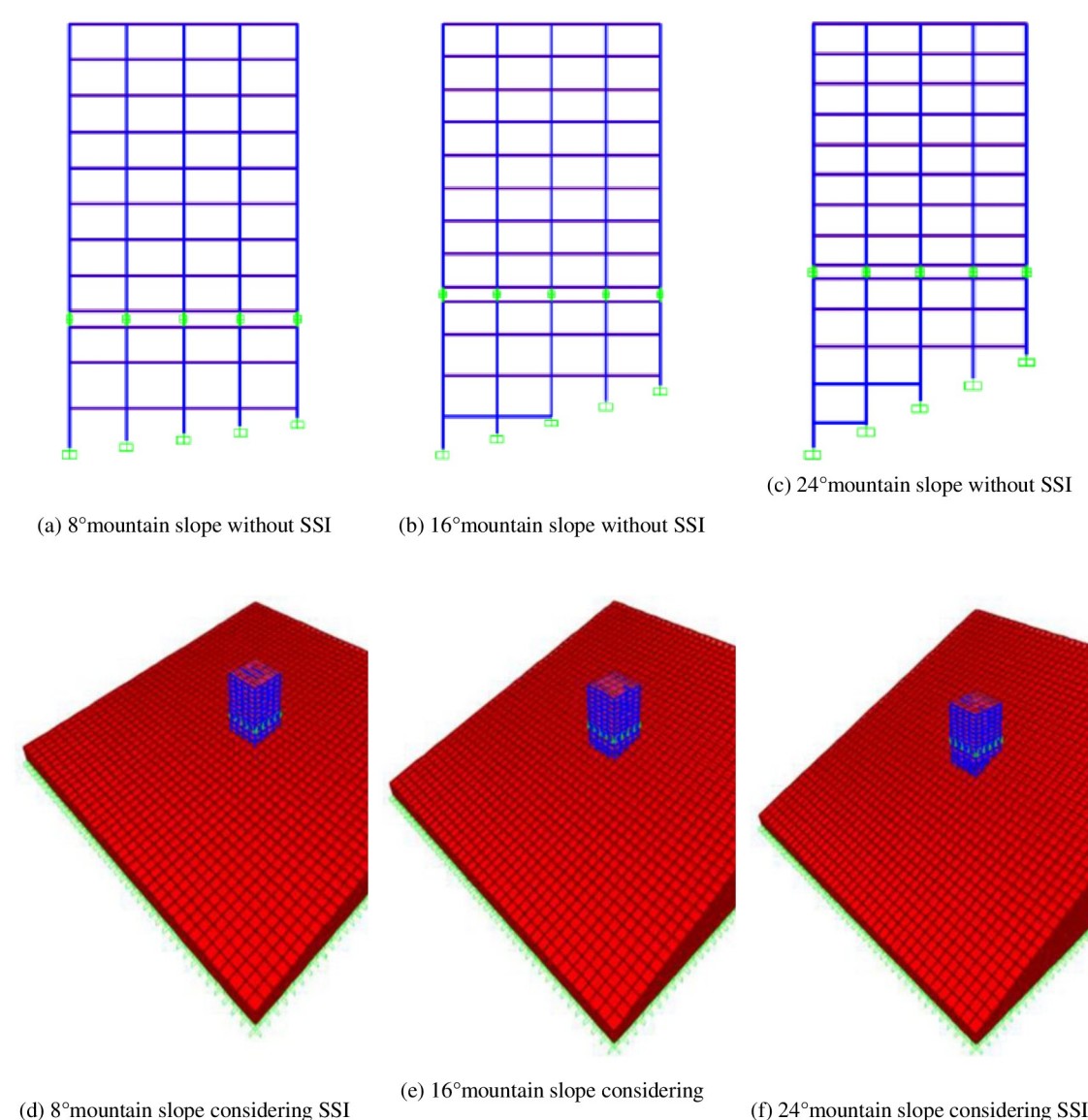

(a) 8°mountain slope without SSI    (b) 16°mountain slope without SSI    (c) 24°mountain slope without SSI

(d) 8°mountain slope considering SSI    (e) 16°mountain slope considering SSI    (f) 24°mountain slope considering SSI

**Fig 4. The schematic diagram of the mid-story isolated stilted structure in mountainous areas considering and without SSI.**

**Table 2. Parameters of isolated bearings.**

| Model number | Effective diameter (mm) | Total rubber thickness (mm) | Stiffness before yielding (kN/m) | 100% equivalent horizontal stiffness (kN/m) | 250% equivalent horizontal stiffness (kN/m) | Vertical stiffness (kN/mm) | Yield force (kN) |
|---|---|---|---|---|---|---|---|
| LRB800 | 800 | 160 | 13808 | 2746 | 1170 | 4355 | 167.5 |
| LRB900 | 900 | 162 | 17046 | 3433 | 2213 | 5233 | 212 |

**Table 3. Parameters of foundation soil.**

| Density (kg·m$^{-3}$) | Poisson ratio | Elastic modulus (MPa) | Shear modulus of elasticity (MPa) | The angle of internal friction (°) | Expansion angle (°) | Shear wave speed (m/s) |
|---|---|---|---|---|---|---|
| 2000 | 0.3 | 325 | 125 | 30 | 18 | 250 |

areas is slightly larger than that of the structure without SSI. The structural period of the mid-story isolated structure considering SSI and without SSI in mountainous areas increases with the increase of mountain slope.

## 4.2 Comparative analysis of inter-story shear force

Inter-story shear can reflect the dynamic response of each layer of the mid-story isolated stilted structures in mountainous areas under earthquakes, and the inter-story shear of seven different mid-story isolated stilted structures as shown in Figs 6 and 7.

As the mountain slope increases, the inter-story shear of the mid-story isolated stilted structure considering SSI in mountainous areas also increases, while the inter-story shear of the mid-story isolated stilted structure without SSI is less affected by the mountain slope. When the mountain slope is 8°, the inter-story shear of the mid-story isolated stilted structure considering SSI in mountainous areas does not increase significantly compared with that of the structure without SSI. When the mountain slope is 24°, the inter-story shear of the mid-story isolated stilted structure considering SSI increases significantly compared with that of the structure without SSI. It is consistent with the increase of parameters such as inter-story displacement and inter-story shear of the structure after considering SSI effect as mentioned in

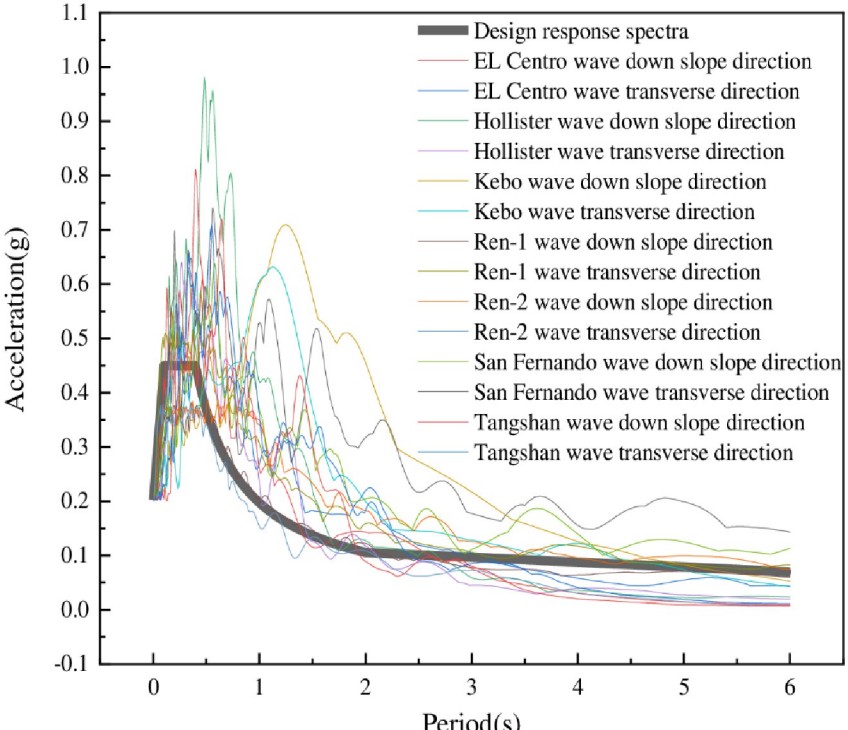

**Fig 5. Acceleration response spectrum of ground motions.**

**Table 4. Parameters of ground motions.**

| Ground motion name | Date | Duration | PGA(m/s²) | Magnitude |
|---|---|---|---|---|
| EL Centro | 1949 | 51 | 0.341(down-slope direction) | 6.9 |
| | | | 0.210(transverse-slope direction) | |
| Hollister | 1961 | 40 | 0.059(down-slope direction) | 5.6 |
| | | | 0.115(transverse-slope direction) | |
| Kobe | 1995 | 42 | 0.315(down-slope direction) | 6.9 |
| | | | 0.278(transverse-slope direction) | |
| San Fernando | 1971 | 28 | 0.059(down-slope direction) | 6.61 |
| | | | 0.044(transverse-slope direction) | |
| Tangshan | 1976 | 49 | 0.065(down-slope direction) | 7.8 |
| | | | 0.055(transverse-slope direction) | |
| Ren-1 | / | 30 | 0.100(down-slope direction) | / |
| | | | 0.100(transverse-slope direction) | |
| Ren-2 | / | 50 | 0.100(down-slope direction) | / |
| | | | 0.100(transverse-slope direction) | |

the papers of Cheng et al. [30], Jin et al. [31], Galal et al. [16] and Zhuang et al. [17]. The inter-story shear of the structures in mountainous areas is influenced by the mountain slope, while the Code for Seismic Design of Buildings (GB 50011–2010) also mentions that when there are unfavorable conditions such as mountainous areas and tilting, the amplification coefficient is considered when analyzing the seismic response of the structure.

The base shear and inter-story shear are influenced by the slope to a similar extent. When considering the SSI effect, the base shear shows a noticeable increase with the increase in the slope of the mountainous terrain. However, when not considering the SSI effect, an increase in the slope of the mountainous terrain does not lead to a significant increase in base shear. This indicates that when considering the SSI in mountainous areas, the slope soil cannot be regarded as a rigid body. As the slope increases, the volume of soil under the structure also increases, and the mutual propagation and exchange effect of vibration energy increases.

The amplification coefficients of the inter-story shear in the down and transverse-slope directions as shown in Figs 8 and 9, it can be seen from that the amplification coefficients of the inter-story shear are concentrated around 1 as the slope increases for the mid-story isolated stilted structure without considering SSI. The amplification coefficients of inter-story shear are concentrated in the range of 1.1 to 1.8 with the increase of the slope for the mid-story isolated stilted structure with SSI in mountainous areas. The amplification coefficient for the mid-story isolated stilted structure considering SSI with a slope ratio of 24°/16° is less than 16°/8°. It further shows that it is inaccurate to disregard SSI effect in practical engineering.

**Table 5. Structural period.**

| Modal | 8° mountain slope without considering SSI (s) | 8° mountain slope considering SSI (s) | 16° mountain slope without considering SSI (s) | 16° mountain slope considering SSI (s) | 24° mountain slope without considering SSI (s) | 24° mountain slope considering SSI (s) |
|---|---|---|---|---|---|---|
| 1 | 2.203 | 2.481 | 2.419 | 2.488 | 2.460 | 2.491 |
| 2 | 2.191 | 2.471 | 2.415 | 2.474 | 2.449 | 2.476 |
| 3 | 1.882 | 2.190 | 2.044 | 2.195 | 2.044 | 2.195 |

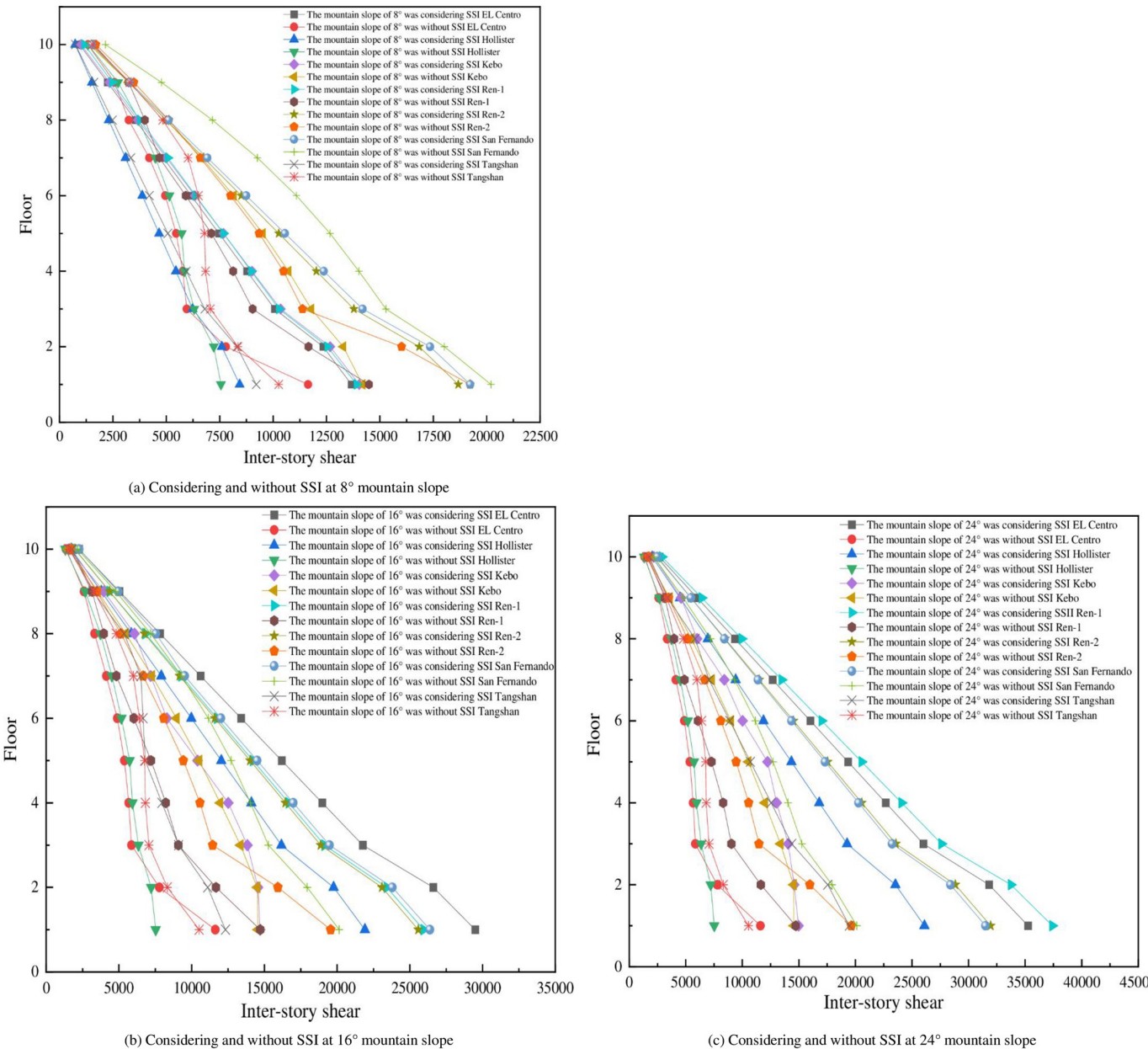

(a) Considering and without SSI at 8° mountain slope

(b) Considering and without SSI at 16° mountain slope

(c) Considering and without SSI at 24° mountain slope

**Fig 6. Inter-story shear of the structure in different down-slope directions.**

### 4.3 The time-history curve of top floor acceleration

The acceleration value of the top floor under earthquakes of the six different mid-story isolated stilted structures was extracted, and the time-history curve as shown in Fig 10.

The peak acceleration time-history curves of the mid-story isolated stilted structure considering SSI in mountainous areas under different mountain slopes as shown in Fig 10a. It can be seen from that with the increase of mountain slope, the peak acceleration of the mid-story isolated stilted structure sharply increases.

The peak acceleration time-history curves of the mid-story isolated stilted structure without SSI under different mountain slopes as shown in Fig 10b. It can be seen from Fig 10b that the

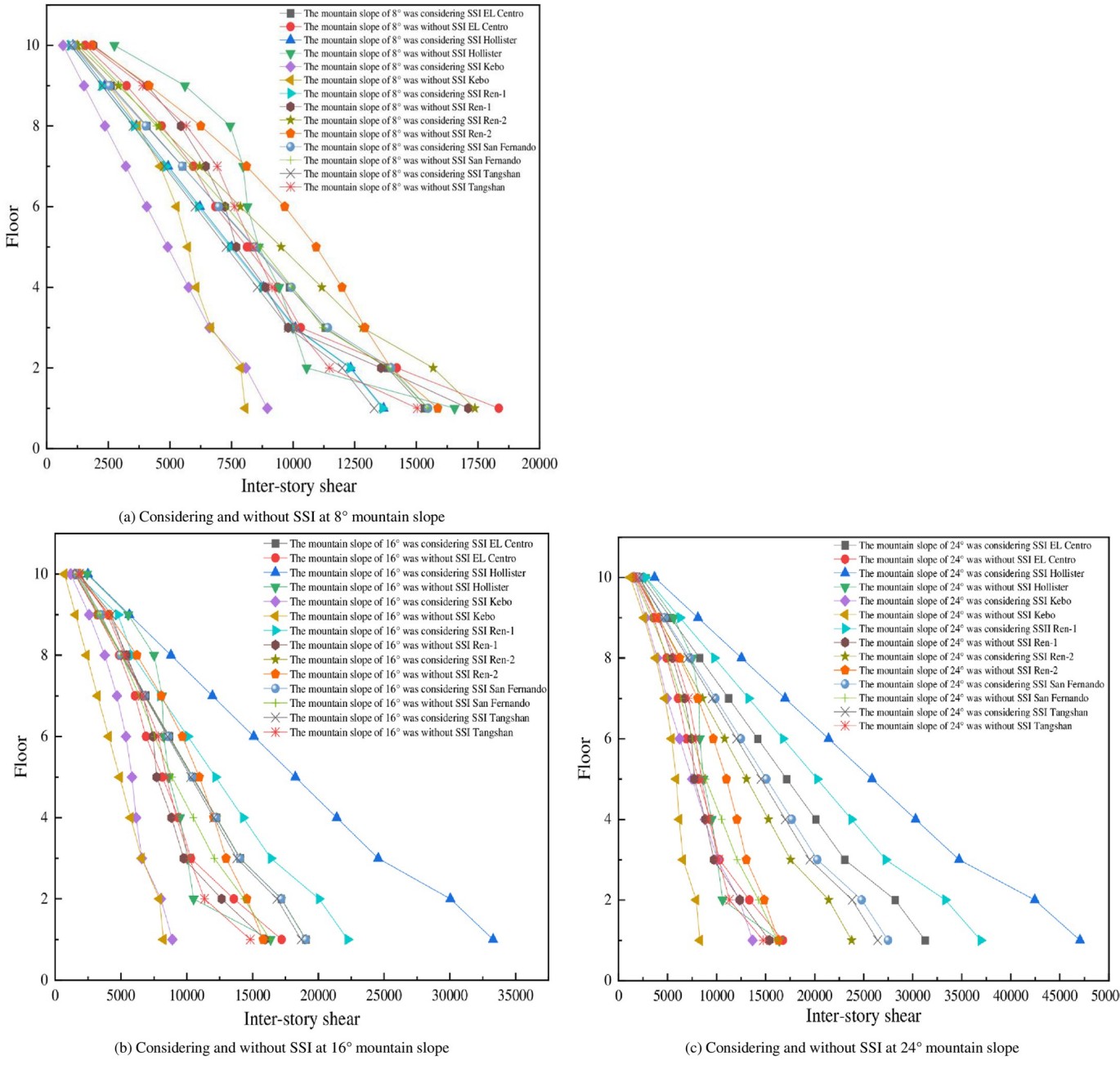

(a) Considering and without SSI at 8° mountain slope

(b) Considering and without SSI at 16° mountain slope

(c) Considering and without SSI at 24° mountain slope

**Fig 7. Inter-story shear of the structure in different transverse-slope directions.**

peak acceleration of the mid-story isolated stilted structure increases insignificantly as the mountain slope increases.

By comparing Fig 10a and 10b, when the mountain slope is 8°, the time-history curve of the top floor acceleration of the mid-story isolated stilted structure considering SSI is close to that of the structure without SSI, and the peak accelerations of the two structures are not much different. As the mountain slope increases, the acceleration time- history curve of the mid-story isolated stilted structure considering SSI is higher than that of the structure without SSI. When

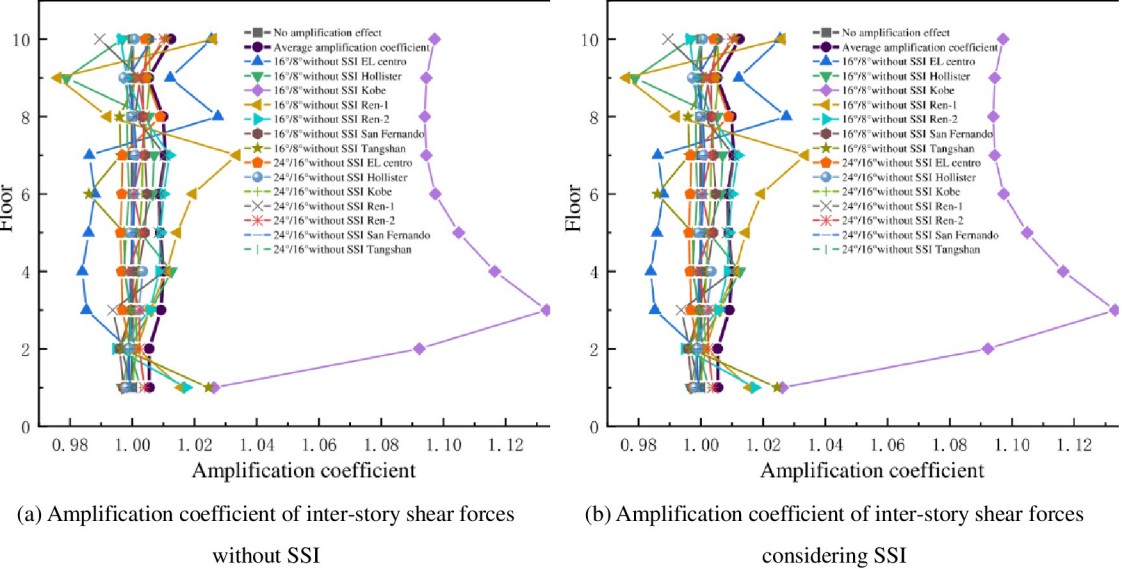

(a) Amplification coefficient of inter-story shear forces

without SSI

(b) Amplification coefficient of inter-story shear forces

considering SSI

**Fig 8. Amplification coefficient of inter-story shear forces in down-slope direction.**

the mountain slope increases to 24°, the peak acceleration of the mid-story isolated stilted structure considering SSI is about 4 times that of the structure without SSI. Compared with the mid-story isolated stilted structure without SSI, the peak acceleration of the top floor of the structure considering SSI in mountainous areas appears later.

### 4.4 Torsion angle

The irregular shape of the structure can lead to torsion during earthquakes, and the presence of torsion can exacerbate structural damage. Therefore, this paper introduces the concept of

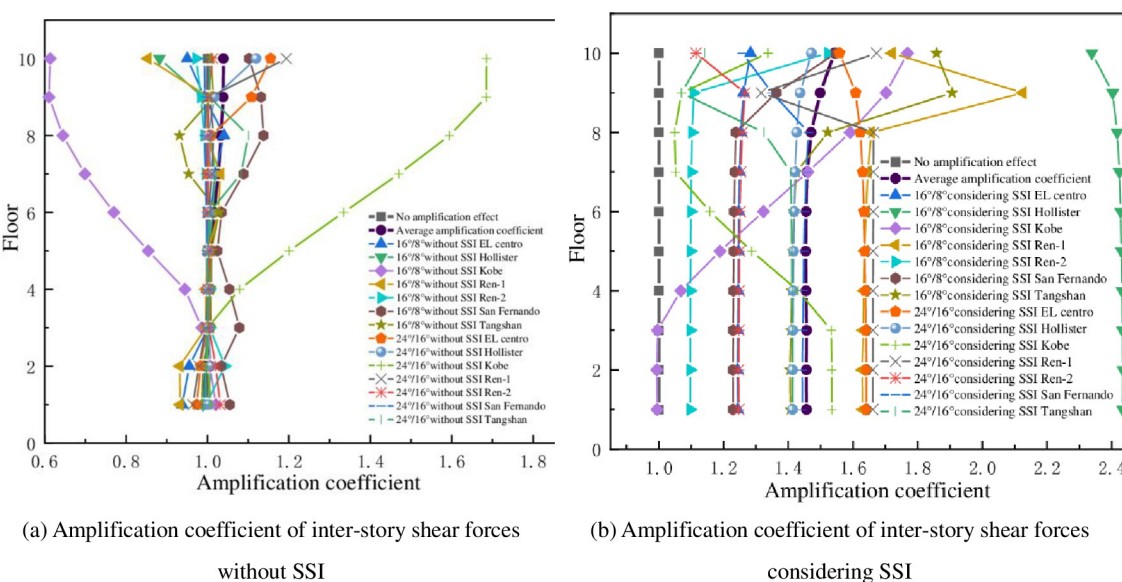

(a) Amplification coefficient of inter-story shear forces

without SSI

(b) Amplification coefficient of inter-story shear forces

considering SSI

**Fig 9. Amplification coefficient of inter-story shear forces in transverse-slope direction.**

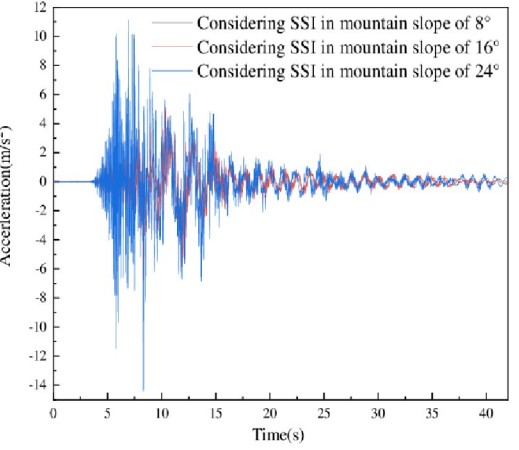
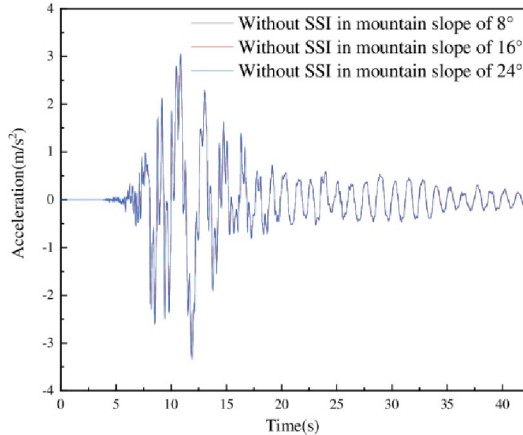

(a) The mid-story isolated stilted structure considering SSI at different mountain slopes

(b) The mid-story isolated stilted structure without SSI at different mountain slopes

**Fig 10.  The peak acceleration time-history curve of the mid-story isolated stilted structure in mountainous areas considering and without SSI.**

torsion angle for quantitative analysis, referring to the concept of torsion angle proposed by Xu Liying et al. in shaking table test [32]. In this paper, the concept of torsion angle is also proposed in the finite element simulation, and the schematic diagram of torsion angle is shown in Fig 11. The structure is asymmetric in transverse-slope direction, that is, there is serious torsion. The torsion angle of the mid-story isolated stilted structure considering SSI in mountainous areas and the structure without SSI in the transverse-slope direction is shown in Fig 12.

$$\theta = \frac{\Delta_1 - \Delta_2}{l}$$

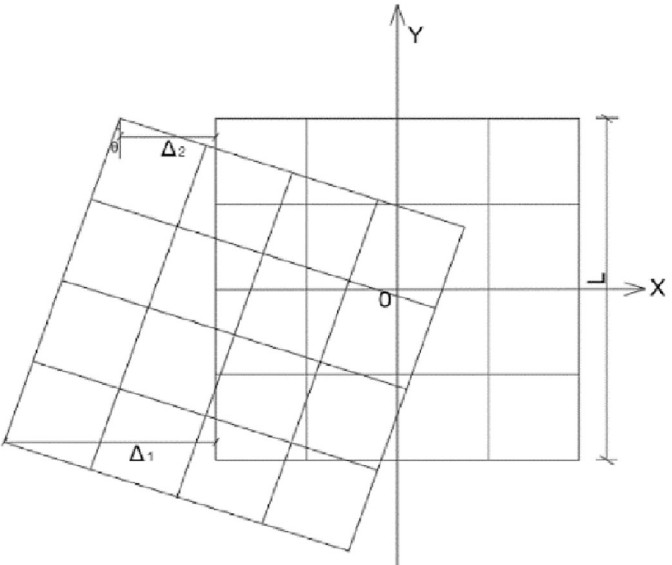

**Fig 11.  Schematic diagram of torsion angle.**

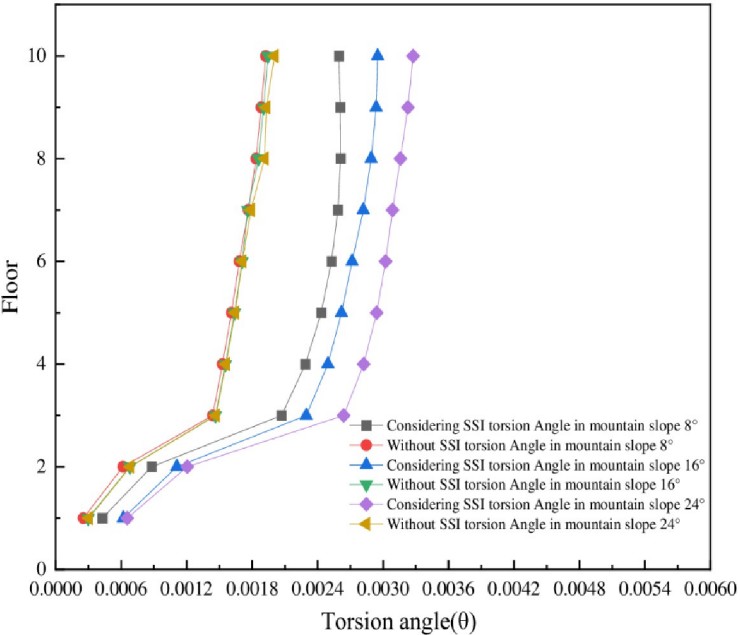

**Fig 12. Torsional angles in transverse-slope direction of six models of the mid-story isolated stilted structure.**

Where $\Delta_1$, $\Delta_2$ is the displacement of the corner column relative to the initial structure, $l$ is the distance between the corresponding corner columns of the structure, which is taken as 24 m in this paper.

The torsion angles in transverse-slope direction of the mid-story isolated stilted structure considering SSI in mountainous areas and the structure without SSI as shown in Fig 12. The torsion angles are represented by their envelope values under seven ground motions. It can be seen from that the torsion angles of the mid-story isolated stilted structure considering SSI in mountainous areas are larger than the structure without SSI. The torsional angles of the mid-story isolated stilted structure are greatly affected by the mountain slope. In the third layer of the structures, there will be a sudden increase, because the third layer is a seismic isolated layer and the stiffness is small.

## 4.5 Structural stress

The stress diagram of the mid-story isolated stilted structures considering SSI and the structures without SSI in different mountain slopes are shown in Fig 13.

It can be seen from that the maximum stress of the mid-story isolated stilted structures considering SSI in mountainous areas ("the new structure" is used to refer to it in the following text) appears in the shortest column of the stilted layer, indicating that the short columns of the new structure are the most unstable, which is consistent with the situation proposed by Wang Liping et al. that the stilted short column of the stilted layer is the first to be destroyed under earthquakes [33].

The stress in the new structure is greater than that in structures without SSI. To meet the slenderness ratio of the long-stilted column, a connecting beam is added to the long-stilted column, and the length of the long-stilted column is reduced to meet the slenderness ratio of the frame column. When connecting beams are added to the long-stilted columns, the height of the long-stilted columns decreases, the stiffness of the long-stilted columns increases, and the

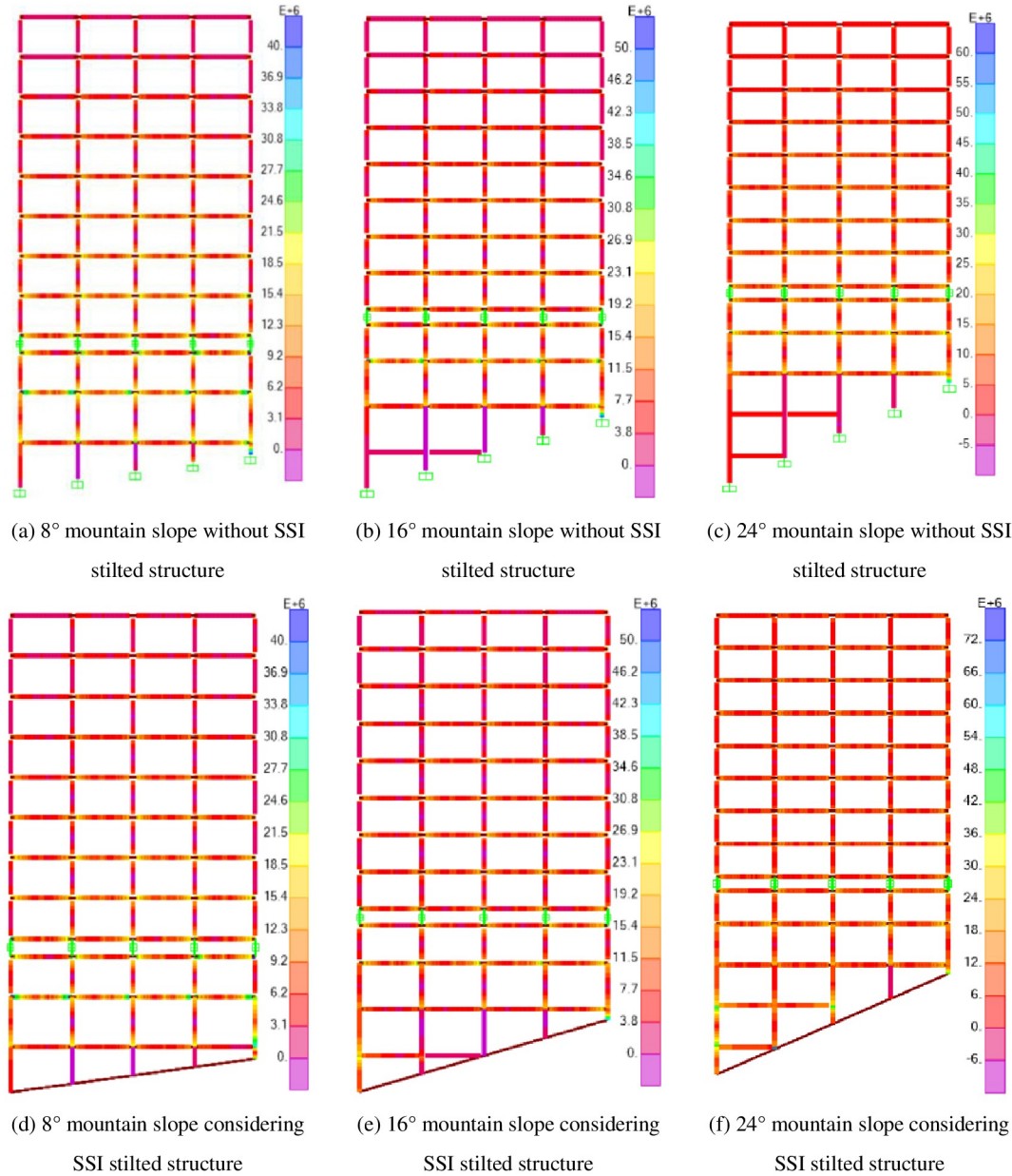

(a) 8° mountain slope without SSI
stilted structure

(b) 16° mountain slope without SSI
stilted structure

(c) 24° mountain slope without SSI
stilted structure

(d) 8° mountain slope considering
SSI stilted structure

(e) 16° mountain slope considering
SSI stilted structure

(f) 24° mountain slope considering
SSI stilted structure

**Fig 13. The stress diagram of six models of the mid-story isolated stilted structures.**

stress concentration is easy to occur. The maximum stress of the stilted layer appears on both sides of the beam and the short-stilted column.

## 5 Foundation earth pressure

For the structure considering SSI effect, earth pressure is also an important evaluation index. The foundation earth pressure diagrams of the mid-story isolated structures considering SSI in different mountain slopes are shown in Fig 14. It can be seen from that with the increase of mountain slope, the foundation earth pressure of the mid-story isolated structures considering SSI in mountainous areas also increases. When the mountain slope is 16°, the stress

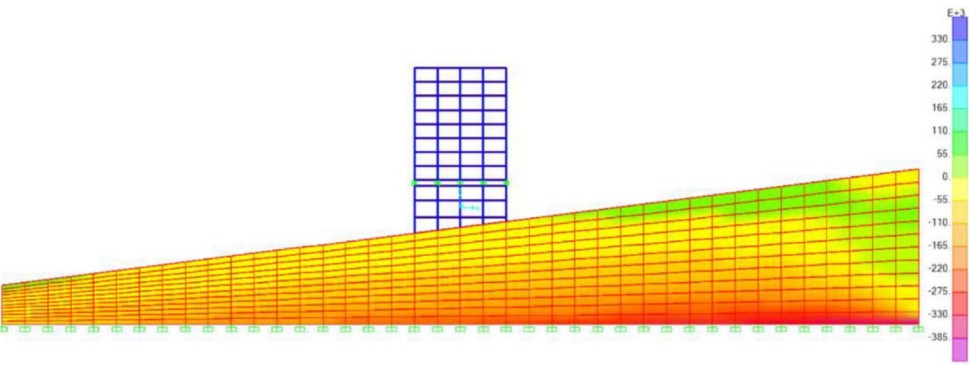

(a) 8° mountain slope considering SSI stilted structure

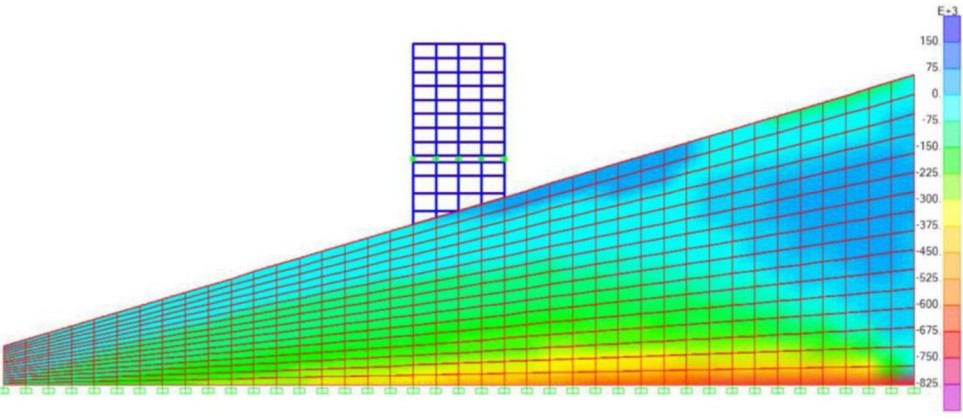

(b) 16° mountain slope considering SSI stilted structure

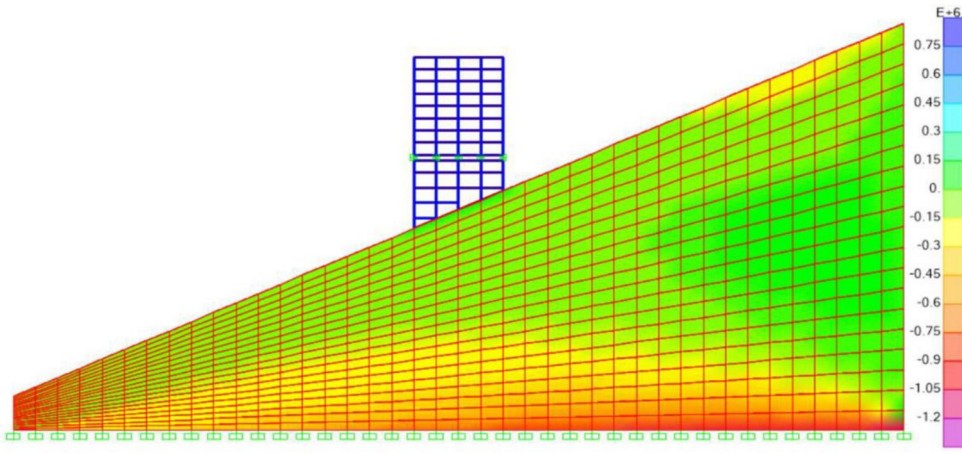

(c) 24° mountain slope considering SSI stilted structure

**Fig 14. The foundation earth pressure diagrams of the mid-story isolated stilted structures considering SSI in different mountain slopes (Unit: N/m2).**

concentration occurs on the foundation of the mid-story isolated stilted structures considering SSI. When the mountain slope is 24°, the stress concentration occurs on the foundation of the mid-story isolated stilted structures considering SSI, which should be paid attention to in engineering.

## 6 Conclusions

Research on the SSI effect on flat sites has yielded many valuable conclusions. However, the current research on the possible impacts of various special local terrains on the dynamic characteristics of structures is still insufficient. For mountainous areas, the seismic performance of this type of structural form can be better revealed through the analysis of the combined action of the upper and lower parts, and considering the influence of slope and SSI effects will bring us closer to the actual seismic effects. In this paper, six models are established under the slope of 8°, 16° and 24°, respectively. The elastic-plastic time-history analysis is carried out. Comparing the seismic response of the new structure and the mid-story isolated stilted structure without SSI in different mountain slopes, the following conclusions are drawn:

1. Compared with not considering SSI, the seismic response of the mid-story isolated stilted structure considering SSI in mountainous areas is amplified and greatly affected by mountain slope. This is because when considering the SSI in mountainous areas, the slope soil cannot be regarded as a rigid body. As the slope increases, the volume of soil under the structure also increases, and the mutual propagation and exchange effect of vibration energy increases.

2. With the increase of mountain slope, the inter-story shear force, torsional angle in the transverse-slope direction, strain and stress of the mid-story isolated stilted structures considering SSI increase, and the seismic response increases, the amplification coefficients are between 1–1.8. However, for structures without SSI, amplification coefficients are concentrated around 1. In this case, the slope soil is regarded as a rigid body, and the increase of the slope has little impact on the propagation of vibration energy, which is less influenced by the slope.

3. With the increase in mountain slope, the foundation earth pressure increases and the stress concentration occurs. The damage to the stilted layer is primarily concentrated at the column end and the beam end, and the maximum seismic response of the stilted layer appears in the short columns.

4. This paper conducts seismic analysis on the mid-story isolated stilted structures considering SSI in mountainous areas. It explores the possible impacts of local terrains in mountainous areas on the dynamic characteristics of structures. This provides a reference for the construction of multi-step, climbing, and laterally staggered floor buildings in mountainous areas.

## Author Contributions

**Formal analysis:** Dewen Liu.

**Methodology:** Jianhua Li.

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
