## [Decision Letter · Decision Letter 0]

24 Jul 2024

PONE-D-24-16917Seismic Response of a Mid-story Isolated Stilted Structure in Mountainous AreasPLOS ONE

Dear Dr. Liu,

Thank you for submitting your manuscript to PLOS ONE. After careful consideration, we feel that it has merit but does not fully meet PLOS ONE’s publication criteria as it currently stands. Therefore, we invite you to submit a revised version of the manuscript that addresses the points raised during the review process.The manuscript written English and fashion does not meet the standard of PLOS ONE. Please **consult professional proofreading services** to improve the English of the manuscript.There are several editorial mistakes, please correct them.Please increase the legend size of the figures. It is hard to read them.Please highlight the design response spectra in Figure 4.Please discuss the results of base shear.Please improve the conclusions and write down them point-wise for ease of readers.Please address the comments specifically and in detail in the response to the reviewer report. Also, do **identify** where in the text you have made the changes by red color, it is particularly helpful to note the page and line numbers from the original manuscript and revision so comparisons can be made. Please submit your revised manuscript by Sep 07 2024 11:59PM. If you will need more time than this to complete your revisions, please reply to this message or contact the journal office at plosone@plos.org. Please include the following items when submitting your revised manuscript:A rebuttal letter that responds to each point raised by the academic editor and reviewer(s). You should upload this letter as a separate file labeled 'Response to Reviewers'.A marked-up copy of your manuscript that highlights changes made to the original version. You should upload this as a separate file labeled 'Revised Manuscript with Track Changes'.An unmarked version of your revised paper without tracked changes. You should upload this as a separate file labeled 'Manuscript'.

We look forward to receiving your revised manuscript.

Kind regards,

Ahad Javanmardi, Ph.D

Academic Editor

PLOS ONE

Journal Requirements:

4. We note that your Data Availability Statement is currently as follows: All relevant data are within the manuscript and its Supporting Information files."

Reviewers' comments:

Reviewer's Responses to Questions

**Comments to the Author**

1. Is the manuscript technically sound, and do the data support the conclusions?

Reviewer #1: Yes

Reviewer #2: Yes

Reviewer #3: Yes

Reviewer #4: Partly

Reviewer #5: Yes

2. Has the statistical analysis been performed appropriately and rigorously? 

Reviewer #1: Yes

Reviewer #2: No

Reviewer #3: Yes

Reviewer #4: No

Reviewer #5: I Don't Know

3. Have the authors made all data underlying the findings in their manuscript fully available?

Reviewer #1: Yes

Reviewer #2: Yes

Reviewer #3: Yes

Reviewer #4: Yes

Reviewer #5: Yes

4. Is the manuscript presented in an intelligible fashion and written in standard English?

Reviewer #1: Yes

Reviewer #2: Yes

Reviewer #3: Yes

Reviewer #4: Yes

Reviewer #5: No

5. Review Comments to the Author

Reviewer #1: In this paper, damage factors of a 10-story midstory isolated stilted building is identified under earthquakes. Six cases differing in the slope value and foundation type are considered for nonlinear dynamic analysis. It is shown that the seismic response is amplified with respect to when SSI is not considered. Increase of the base slope is shown to have an increasing effect on the SSI responses up to 80% while it does not affect the rigid-base responses. The foundation soil stresses increase in this case as well. At the stilted level, the maximum seismic response is shown to be belonging to the short columns.

The paper needs a major revision based on the following:

1. First of all, there is several and several times of repeating “mid-story isolated stilted structures considering SSI in mountainous areas”. It is not necessary. Please resort to a brief expression after the first use of the complete phrase.

2. At the Introduction, please clearly introduce the “mid-story isolated stilted structure” and mention its purpose. It remains to the end of the paper.

3. “The seismic precautionary intensity is VIII, the design basic seismic acceleration value is 0.20 g, the construction site classification is II, the design seismic grouping is the second group”: Please define all of these expressions.

4. “the isolated layer is set at the bottom of the column on the third floor”: What is meant by the isolation layer? Why is not it placed at the foundation level?

5. Please give the mechanical properties of the so-called “HRB400” and “HPB300” steel.

6. What is meant by this sentence: “The ground motions are input separately in down slope and transverse slopes”? The input pattern is unknown. If the input motion is the same along the slope, where is the effect of spatial variation of ground motion in a ground with changing depth?

7. “the peak value of ground motions under rare earthquakes is adjusted to 400 cm/s2”: This is an outdated method of ground motion scaling. It should be based on spectral amplitude or the frequency content.

8. Fig. 5: The selected ground motions are not consistent. There is a large discrepancy between the spectral values.

6. “With the increase of mountain slope, the inter-story shear of the mid-story isolated stilted structure considering SSI in mountainous areas increases, while the inter-story shear of the mid-story isolated stilted structure without SSI is less affected by the mountain slope”: What is expected to be the reason for this phenomenon?

7. “The irregular shape of the structure will lead to torsion under earthquakes … The structure is asymmetric in transverse slope, that is, there is serious torsion”: In this study, it happens only when the ground motion is perpendicular to the stepped frame. It is not clear whether this point was considered or not.

Reviewer #2: In this paper, the seismic response of a mid-story isolated stilted structure in mountainous area considering soil structure interaction has been investigated. Generally, this paper is very interesting and well-written. This reviewer thinks after a round of revision, it can be accepted.

- Abstract should be improved.

-SSI is the main contribution of this paper, some references about SSI are absent and should be added. Optimum tuned tandem mass dampers for suppressing seismic-induced vibrations considering soil-structure interaction. Structures 2023;52:1146-1159. Effect of foundation embedment ratio in suppressing seismic-induced vibrations using optimum tuned mass damper. Soil Dynamics and Earthquake Engineering 2023;171: 107981. Seismic Structure-Soil-Structure Interaction between inelastic structures. Earthquake Engng Struct Dyn. 2024;53:1446–1464.

- I think the structural model is analyzed using sap2000. Detailed information on how the structure-soil interaction system is constructed (which element was used, where the soil properties were defined, etc.) should be added.

-In addition to PGA, PGV also has a significant effect on structural responses (i.e, https://doi.org/10.1007/s40430-022-03895-z). A discussion in section 3.3, considering the above-mentioned article, would be extremely useful for researchers.

- Conclusion section should be improved.

Reviewer #3: Comments:

1. Please check the manuscript for grammatical errors.

2. Please add the basis for selecting seismic motions in Section 2.3.

3. Please explain the meaning of "down slope" and "transverse slope", and explain the main differences between them in the manuscript.

4. In Table 1, the cross-sectional dimensions in mm of the frame columns are 600x600 and of main beams 700x350 and of the secondary beams 600×300. Usually, the first number is the width and the second is the depth of the member. Is the first number here width or depth? Please clarify.

5. In Section 4.2, "and the inter-story shear of six different mid-story isolated stilted structures as shown in Fig. 5 and Fig. 6" should be changed to "and the inter-story shear of six different mid-story isolated stilted structures as shown in Fig. 5 and Fig. 6". There are many similar errors in the manuscript. Please check and modify them.

6. The legends in Figures 4, 5, 6, 7, 8 and 11 are too small to see clearly. Please improve them.

Reviewer #4: It's interesting that this manuscript focuses on seismic response of a mid-story isolated stilted structure in mountainous areas.

1. What is the amount of reinforcement for beams and columns, and the size and quantity of reinforcement used, which may affect the seismic response of structure, should be clearly explained.

2. The floor plan of the structure is not introduced, and the layout of the structure may affect the seismic response of structure, so it should be clearly explained in these aspects.

3.Some of the research addressing these issues should be acknowledged, some recommended references, among many others are,

https://doi.org/10.1016/j.istruc.2020.12.089. https://doi.org/10.3390/app7121238. https://doi.org/10.3311/PPci.15276.

As presented, the writing is not acceptable for the journal. There are problems with sentence structure, verb tense, and clause construction, the quality of English needs improving.

It is my opinion that the paper can be accepted in the journal by providing some revisions mainly devoted to improve its quality and readability.?

Reviewer #5: This paper adopted the numerical software to analyze the seismic response of a mid-story isolated stilted structure models considering SSI. Some interesting results were obtained. But this manuscript can't be accepted in the present form. The authors should address the following comments well before the publication of this paper.

1. Question 1: The authors should highlight the main contribution and advantages compared with the existing literatures.

2. Question 2: Which FE software was used in the study?

3. Question 3: It should be noted that abbreviations that first appear in the manuscript must be given their full names, such as “LRB”.

4. Question 4: It is not clear which constitutive model of soil was adopted in the simulation.

5. Question 5: In the numerical simulation, the reinforcement details should be provided.

6. Question 6: The size of the soil elements should be presented explicitly, including shear wave velocity of site soil and the maximum vibration frequency of earthquake waves.

7. Question 7: During the parametric study, authors should explain why the maximum seismic response increases insignificantly as the mountain slope increases. Please explain the mechanism.

8. Question 8: The word “Kebo” should be revised as “Kobe”. In section 4.2, there should be a space in “andFig. 6”. Please correct the relevant errors in the full text carefully.

9. Question 9: The conclusion is weak and unclear. Additionally, the contribution and novelty are not clearly indicated.

10. Question 10: English should be improved. Unprofessional words should be avoided.

6. PLOS authors have the option to publish the peer review history of their article (what does this mean?). If published, this will include your full peer review and any attached files.

Reviewer #1: No

Reviewer #2: No

Reviewer #3: No

Reviewer #4: No

Reviewer #5: No

---

## [Author Response · Author response to Decision Letter 0]

29 Aug 2024

Response to reviewers

Dear Editor,

Thanks for your letter and thanks for the review comments concerning our manuscript entitled “Seismic Response of a Mid-story Isolated Stilted Structure in Mountainous Areas”. The comments are all valuable and helpful for revising and improving our paper. We have studied all comments carefully and have made conscientious corrections. Modified portions are marked in red on the paper. The main corrections to the paper and the responses to the review comments are listed as follows: 

Below are our detailed responses to all the comments.

Reply to Reviewer 1

Dear Professor,

Thank you for giving us the opportunity to revise our paper. We have read your comments carefully and revised our paper. Thank you very much for your valuable comments, which gave me a better understanding of this research.

Comment 1: First of all, there is several and several times of repeating “mid-story isolated stilted structures considering SSI in mountainous areas”. It is not necessary. Please resort to a brief expression after the first use of the complete phrase.

Response 1: We appreciate the valuable comments and suggestions provided by the reviewer. These comments have played a constructive role in refining the content and structure of the paper. We have made modifications in Section 4.5. After its first appearance, "the new structure" is used to refer to it in the following text.

Comment 2: At the Introduction, please clearly introduce the “mid-story isolated stilted structure” and mention its purpose. It remains to the end of the paper.

Response 2: Thank you very much for your valuable comments. We have made modifications at the Introduction. The mid-story isolated stilted structures are stilted structures that adopt mid-story isolation. Mid-story isolated structure is a new type of isolated structure developed from base isolated structure, in which the isolation device is installed between the upper and lower stories of a building or set up between the substructure and the main structure, to control the earthquake response of the structure.

Comment 3: “The seismic precautionary intensity is VIII, the design basic seismic acceleration value is 0.20 g, the construction site classification is II, the design seismic grouping is the second group”: Please define all of these expressions.

Response 3: Thank you very much for your valuable comments. We have made modifications. According to the Code for Seismic Design of Buildings (GB 50011-2010), earthquake intensity is a measure of the degree of influence of an earthquake on the surface and engineering buildings. It is divided into 12 grades, with grade 1 being the weakest and grade 12 being the strongest. The seismic precautionary intensity is VIII, which means that the building can resist the damage of an earthquake intensity of 8 degrees. The design basic seismic acceleration value of 0.20 g means that the design value of the seismic acceleration with an exceeding probability of 10% within 50 years in this area is 0.20 g. The building site category is a classification of building sites based on the thickness of the site overburden and the shear wave velocity of the soil layer. In this study, the shear wave velocity of the soil layer is 250 m/s, the overburden thickness is 10 m, and the site category is level II. The design earthquake grouping is a parameter used to characterize the influence of earthquake magnitude and epicentral distance. It is divided into the first group (near earthquake area), the second group (medium and far earthquake area), and the third group (far earthquake area).

Comment 4: “the isolated layer is set at the bottom of the column on the third floor”: What is meant by the isolation layer? Why is not it placed at the foundation level?

Response 4: Thank you very much for your valuable comments. We have made modifications. The isolated layer is the floor where isolation devices are installed. Due to the fact that it is not suitable to set the isolated layer at the foundation level for the stilted building structure in mountainous areas, the isolated layer is set at the bottom of the columns on the third floor.

Comment 5: Please give the mechanical properties of the so-called “HRB400” and “HPB300” steel.

Response 5: Thank you very much for your valuable comments. We have added the mechanical properties of HRB400 steel and HPB300 steel in Section 3.1: the specified yield strength of HRB400 steel was f_y=400MP_a; the specified yield strength of HPB300 steel was f_y=300MP_a.

Comment 6: What is meant by this sentence: “The ground motions are input separately in down slope and transverse slopes”? The input pattern is unknown. If the input motion is the same along the slope, where is the effect of spatial variation of ground motion in a ground with changing depth?

Response 6: Thank you very much for your valuable comments. We have made modifications in the main text. We changed “down slope” to “down-slope direction” and “transverse slope” to “transverse-slope direction”. Among them, the down-slope direction is parallel to the slope direction, and the transverse-slope direction is perpendicular to the slope direction.

Comment 7: “the peak value of ground motions under rare earthquakes is adjusted to 400 cm/s2”: This is an outdated method of ground motion scaling. It should be based on spectral amplitude or the frequency content.

Response 7: Thank you very much for your valuable comments. We have made modifications in Section 3.3.

Comment 8: Fig. 5: The selected ground motions are not consistent. There is a large discrepancy between the spectral values.

Response 8: Thank you very much for your valuable comments. We have made modifications. There is a certain gap in magnitude and peak ground acceleration between the ground motions selected in this study. Therefore, there are great differences in inter-story shear in Fig 5(now renamed Fig. 6).

Comment 9: “With the increase of mountain slope, the inter-story shear of the mid-story isolated stilted structure considering SSI in mountainous areas increases, while the inter-story shear of the mid-story isolated stilted structure without SSI is less affected by the mountain slope”: What is expected to be the reason for this phenomenon?

Response 9: Thank you very much for your valuable comments. We have made modifications. This phenomenon may be because when the SSI in mountainous areas is not considered, the slope soil is regarded as a rigid body, and the increase in slope has little influence on the propagation of vibration energy. However, when considering the SSI in mountainous areas, the slope soil cannot be regarded as a rigid body. As the slope increases, the volume of soil under the structure also increases, and the mutual propagation and exchange effect of vibration energy increases.

Comment 10: “The irregular shape of the structure will lead to torsion under earthquakes … The structure is asymmetric in transverse slope, that is, there is serious torsion”: In this study, it happens only when the ground motion is perpendicular to the stepped frame. It is not clear whether this point was considered or not.

Response 10: Thank you very much for your valuable comments. We have made modifications. This study considers three-directional seismic action and includes this point.

Thank you again for your valuable comments. I am looking forward to hearing from you soon.

Yours Sincerely,

Dewen Liu

Reply to Reviewer 2

Dear Professor,

Thank you for giving us the opportunity to revise the paper. We have read your comments carefully and have revised our paper. Thank you very much for your valuable comments, which gave me a better understanding of this research.

Comment 1: Abstract should be improved.

Response 1: Thank you very much for your valuable comments. We have improved the abstract. 

Comment 2: SSI is the main contribution of this paper, some references about SSI are absent and should be added. Optimum tuned tandem mass dampers for suppressing seismic-induced vibrations considering soil-structure interaction. Structures 2023;52:1146-1159. Effect of foundation embedment ratio in suppressing seismic-induced vibrations using optimum tuned mass damper. Soil Dynamics and Earthquake Engineering 2023;171: 107981. Seismic Structure-Soil-Structure Interaction between inelastic structures. Earthquake Engng Struct Dyn. 2024;53:1446–1464.

Response 2: Thank you very much for your valuable comments. We have cited all the references recommended by the reviewer. Section 1 references are as follows:

Onur A, Ehsan F. Optimum tuned tandem mass dampers for suppressing seismic-induced vibrations considering soil-structure interaction. Structures,52,1146-1159(2023). 

Onur A, Tufan C, Ozturk K, Dilek K. Effect of foundation embedment ratio in suppressing seismic-induced vibrations using optimum tuned mass damper. Soil Dynamics and Earthquake Engineering, 171(2023).

Vicencio F, Alexander N, Málaga-Chuquitaype C. Seismic Structure-Soil-Structure Interaction between inelastic structures. Earthquake Engng Struct Dyn, 53(04), 1446-1464(2024). 

Comment 3: I think the structural model is analyzed using sap2000. Detailed information on how the structure-soil interaction system is constructed (which element was used, where the soil properties were defined, etc.) should be added.

-In addition to PGA, PGV also has a significant effect on structural responses (i.e, https://doi.org/10.1007/s40430-022-03895-z). A discussion in section 3.3, considering the above-mentioned article, would be extremely useful for researchers.

Response 3: Thank you very much for your valuable comments. We have made modifications. In Section 3.2, Table 3 defines soil properties. The elements used include: Density, Poisson Ratio, Elastic modulus, Shear modulus of elasticity, The angle of internal friction, Expansion angle and Shear wave speed. We have cited the references recommended by the reviewer. Section 3.3 references are as follows:

Onur A, Tufan C, Ozturk K. Effect of earthquake frequency content on seismic-induced vibration control of structures equipped with tuned mass damper. Journal of the Brazilian Society of Mechanical Sciences and Engineering, 44, 584(2022). 

Comment 4: Conclusion section should be improved.

Response 4: Thank you very much for your valuable comments. We have improved the conclusion.

Thank you again for your valuable comments. I am looking forward to hearing from you soon.

Yours Sincerely,

Dewen Liu

Reply to Reviewer 3

Dear Professor,

Thank you for giving us the opportunity to revise the paper. We have read your comments carefully and have revised our paper. Thank you very much for your valuable comments, which gave me a better understanding of this research.

Comment 1: Please check the manuscript for grammatical errors.

Response 1: Thank you very much for your valuable comments. We have checked and corrected the discovered grammatical errors.

Comment 2: Please add the basis for selecting seismic motions in Section 3.3.

Response 2: Thank you very much for your valuable comments. We have made modifications to Section 3.3.

Comment 3: Please explain the meaning of "down slope" and "transverse slope", and explain the main differences between them in the manuscript.

Response 3: Thank you very much for your valuable comments. We have made modifications in the main text. We changed “down slope” to “down-slope direction” and “transverse slope” to “transverse-slope direction”. Among them, the down-slope direction is parallel to the slope direction, and the transverse-slope direction is perpendicular to the slope direction.

Comment 4: In Table 1, the cross-sectional dimensions in mm of the frame columns are 600x600 and of main beams 700x350 and of the secondary beams 600×300. Usually, the first number is the width and the second is the depth of the member. Is the first number here width or depth? Please clarify.

Response 4: Thank you very much for your valuable comments. We have added an explanation in Table 1. The first number is the depth and the second number is the width. 

Comment 5: In Section 4.2, "and the inter-story shear of six different mid-story isolated stilted structures as shown in Fig. 5 and Fig. 6" should be changed to "and the inter-story shear of six different mid-story isolated stilted structures as shown in Fig. 5 and Fig. 6". There are many similar errors in the manuscript. Please check and modify them.

Response 5: Thank you very much for your valuable comments. We rechecked the manuscript and corrected the discovered errors.

Comment 6: The legends in Figures 4, 5, 6, 7, 8 and 11 are too small to see clearly. Please improve them.

Response 6: Thank you very much for your valuable comments. We have tried our best to improve the legends. However, considering aesthetics and professionalism, it is difficult to make too much modification. We are very sorry for this.

Thank you again for your valuable comments. I am looking forward to hearing from you soon.

Yours Sincerely,

Dewen Liu

Reply to Reviewer 4

Dear Professor,

Thank you for giving us the opportunity to revise the paper. We have read your comments carefully and have revised our paper. Thank you very much for your valuable comments, which gave me a better understanding of this research.

Comment 1: What is the amount of reinforcement for beams and columns, and the size and quantity of reinforcement used, which may affect the seismic response of structure, should be clearly explained.

Response 1: Thank you very much for your valuable comments. We have made modifications. We have provided the reinforcement of beams and columns in Table 1. Among them, the column is reinforced with 16 steel bars with a diameter of 25 mm, and the beam is reinforced with 5 steel bars with a diameter of 20 mm. 

Comment 2: The floor plan of the structure is not introduced, and the layout of the structure may affect the seismic response of structure, so it should be clearly explained in these aspects.

Response 2: Thank you very much for your valuable comments. We have supplemented Figure 3 "Planar calculation sketch" in Section 3.1.

Comment 3: Some of the research addressing these issues should be acknowledged, some recommended references, among many others are,

https://doi.org/10.1016/j.istruc.2020.12.089. 

https://doi.org/10.3390/app7121238.

https://doi.org/10.3311/PPci.15276.

Response 3: Thank you very much for your valuable comments. We have cited all the references recommended by the reviewer. Section 3.3 references are as follows:

Liu C, Fang D, Zhao L. Reflection on earthquake damage of buildings in 2015 Nepal earthquake and seismic measures for post-earthquake reconstruction. Structures, 30, 647-658(2021).

Liu C, Yang W, Yan Z, Lu Z, Luo N. Base Pounding Model and Response Analysis of Base-Isolated Structures under Earthquake Excitation. Submission received, 7(12), 1238(2017).

Liu C, Fang D, Yan Z. Seismic Fragility Analysis of Base Isolated Structure Subjected to Near-fault Ground Motions. Period. Polytech. Civil Eng. [Internet]. 2021 Jan. 1 [cited 2024 Aug. 22];65(3):768-83.

Thank you again for your valuable comments. I am looking forward to hearing from you soon.

Yours Sincerely,

Dewen Liu

Reply to Reviewer 5

Dear Professor,

Thank you for giving us the opportunity to revise the paper. We have read your comments carefully and have revised our paper. Thank you very much for your valuable comments, which gave me a better understanding of this research.

Comment 1: The authors should highlight the main contribution and advantages compared with the existing literatures.

Response 1: Thank you very much for your valuable comments. We have added a discussion on “the main contribution and advantages compared with the existing literatures” in Section 1.

Comment 2: Which FE software was used in the study?

Response 2: Thank you very much for your valuable comments. In this study, the FE software we use is sap2000. 

Comment 3: It should be noted that abbreviations that first appear in the manuscript must be given their full names, such as “LRB”.

Response 3: Thank you very much for your valuable comments. We have added anno

---

## [Decision Letter · Decision Letter 1]

23 Sep 2024

PONE-D-24-16917R1Seismic Response of a Mid-story Isolated Stilted Structure in Mountainous AreasPLOS ONE

Dear Dr. Liu,

Thank you for submitting your manuscript to PLOS ONE. After careful consideration, we feel that it has merit but does not fully meet PLOS ONE’s publication criteria as it currently stands. Therefore, we invite you to submit a revised version of the manuscript that addresses the points raised during the review process.

 Please address the comments specifically and in detail in the response to the reviewer report. Also, do **identify** where in the text you have made the changes by red color, it is particularly helpful to note the page and line numbers from the original manuscript and revision so comparisons can be made. Please submit your revised manuscript by Nov 07 2024 11:59PM. If you will need more time than this to complete your revisions, please reply to this message or contact the journal office at plosone@plos.org. Please include the following items when submitting your revised manuscript:A rebuttal letter that responds to each point raised by the academic editor and reviewer(s). You should upload this letter as a separate file labeled 'Response to Reviewers'.A marked-up copy of your manuscript that highlights changes made to the original version. You should upload this as a separate file labeled 'Revised Manuscript with Track Changes'.An unmarked version of your revised paper without tracked changes. You should upload this as a separate file labeled 'Manuscript'.If applicable, we recommend that you deposit your laboratory protocols in protocols.io to enhance the reproducibility of your results. Protocols.io assigns your protocol its own identifier (DOI) so that it can be cited independently in the future. For instructions see: https://journals.plos.org/plosone/s/submission-guidelines#loc-laboratory-protocols. Additionally, PLOS ONE offers an option for publishing peer-reviewed Lab Protocol articles, which describe protocols hosted on protocols.io. Read more information on sharing protocols at https://plos.org/protocols?utm_medium=editorial-email&utm_source=authorletters&utm_campaign=protocols.

We look forward to receiving your revised manuscript.

Kind regards,

Ahad Javanmardi, Ph.D

Academic Editor

PLOS ONE

Journal Requirements:

Additional Editor Comments:The manuscript written English and fashion does not meet the standard of PLOS ONE. Please **consult professional proofreading services** to improve the English of the manuscript.There are several editorial mistakes, please correct them.Please increase the legend size of the figures. It is hard to read them.Please highlight the design response spectra in Figure 4.Please discuss the results of base shear.Please improve the conclusions and write down them point-wise for ease of readers Reviewers' comments:

Reviewer's Responses to Questions

**Comments to the Author**

1. If the authors have adequately addressed your comments raised in a previous round of review and you feel that this manuscript is now acceptable for publication, you may indicate that here to bypass the “Comments to the Author” section, enter your conflict of interest statement in the “Confidential to Editor” section, and submit your "Accept" recommendation.

Reviewer #1: All comments have been addressed

Reviewer #2: All comments have been addressed

Reviewer #3: All comments have been addressed

Reviewer #4: All comments have been addressed

Reviewer #5: All comments have been addressed

2. Is the manuscript technically sound, and do the data support the conclusions?

Reviewer #1: (No Response)

Reviewer #2: Yes

Reviewer #3: Yes

Reviewer #4: Yes

Reviewer #5: Yes

3. Has the statistical analysis been performed appropriately and rigorously? 

Reviewer #1: (No Response)

Reviewer #2: I Don't Know

Reviewer #3: Yes

Reviewer #4: Yes

Reviewer #5: Yes

4. Have the authors made all data underlying the findings in their manuscript fully available?

Reviewer #1: (No Response)

Reviewer #2: Yes

Reviewer #3: Yes

Reviewer #4: Yes

Reviewer #5: Yes

5. Is the manuscript presented in an intelligible fashion and written in standard English?

Reviewer #1: (No Response)

Reviewer #2: Yes

Reviewer #3: Yes

Reviewer #4: Yes

Reviewer #5: Yes

6. Review Comments to the Author

Reviewer #1: (No Response)

Reviewer #2: The authors have made the requested revisions. Therefore, I recommend that the work be published in your journal.

Reviewer #3: (No Response)

Reviewer #4: after revised, the authors have adequately addressed all comments raised in a previous round of review，so，

it can be accepted in current situation.

Reviewer #5: The manuscript is technically sound, and do the data support the conclusions. The authors have made all data underlying the findings in their manuscript fully availableThe paper can be accepted.

7. PLOS authors have the option to publish the peer review history of their article (what does this mean?). If published, this will include your full peer review and any attached files.

Reviewer #1: No

Reviewer #2: No

Reviewer #3: No

Reviewer #4: No

Reviewer #5: No

---

## [Author Response · Author response to Decision Letter 1]

29 Sep 2024

Response to reviewers

Dear Editor,

Thanks for your letter and thanks for the review comments concerning our manuscript entitled “Seismic Response of a Mid-story Isolated Stilted Structure in Mountainous Areas”. The comments are all valuable and helpful for revising and improving our paper. We have studied all comments carefully and have made conscientious corrections. Modified portions are marked in red on the paper. The main corrections to the paper and the responses to the review comments are listed as follows: 

Below are our detailed responses to all the comments.

Reply to Academic Editor

Dear Professor,

Thank you for giving us the opportunity to revise our paper. We have read your comments carefully and revised our paper. Thank you very much for your valuable comments, which gave me a better understanding of this research.

Comment 1: The manuscript written English and fashion does not meet the standard of PLOS ONE. Please consult professional proofreading services to improve the English of the manuscript.

Response 1: We appreciate the valuable comments and suggestions provided by the academic editor. These comments have played a constructive role in refining the content and structure of the paper. We have improved the English.

Comment 2: There are several editorial mistakes, please correct them.

Response 2: Thank you very much for your valuable comments. We have checked and corrected the discovered editorial mistakes.

Comment 3: Please increase the legend size of the figures. It is hard to read them.

Response 3: Thank you very much for your valuable comments. We have increased the size of the legends for figures 5, 6, 7, 8(a), 9(b), 10, and 12.

Comment 4: Please highlight the design response spectra in Figure 4.

Response 4: Thank you very much for your valuable comments. We have modified figure 5 to highlight the design response spectra.

Comment 5: Please discuss the results of base shear.

Response 5: Thank you very much for your valuable comments. In Section 4.2, we have added a discussion on base shear. Content as follows：

“The base shear and inter-story shear are influenced by the slope to a similar extent. When considering the SSI effect, the base shear shows a noticeable increase with the increase in the slope of the mountainous terrain. However, when not considering the SSI effect, an increase in the slope of the mountainous terrain does not lead to a significant increase in base shear. This indicates that when considering the SSI in mountainous areas, the slope soil cannot be regarded as a rigid body. As the slope increases, the volume of soil under the structure also increases, and the mutual propagation and exchange effect of vibration energy increases.”

Comment 6: Please improve the conclusions and write down them point-wise for ease of readers.

Response 6: Thank you very much for your valuable comments. We have improved the conclusions and written them down point-wise.

Thank you again for your valuable comments. I am looking forward to hearing from you soon.

Yours Sincerely,

Dewen Liu

---

## [Editor Report · Decision Letter 2]

8 Oct 2024

Seismic Response of a Mid-story Isolated Stilted Structure in Mountainous Areas

PONE-D-24-16917R2

Dear Dr. Liu,

We’re pleased to inform you that your manuscript has been judged scientifically suitable for publication and will be formally accepted for publication once it meets all outstanding technical requirements.

Kind regards,

Ahad Javanmardi, Ph.D

Academic Editor

PLOS ONE
---

## [Editor Report · Acceptance letter]

17 Oct 2024

PONE-D-24-16917R2 

PLOS ONE

Dear Dr. Liu, 

I'm pleased to inform you that your manuscript has been deemed suitable for publication in PLOS ONE. Congratulations! Your manuscript is now being handed over to our production team.

Kind regards, 

on behalf of

Associate Professor Ahad Javanmardi 

Academic Editor

PLOS ONE